# Improving Known–Unknown Cattle’s Face Recognition for Smart Livestock Farm Management

**DOI:** 10.3390/ani13223588

**Published:** 2023-11-20

**Authors:** Yao Meng, Sook Yoon, Shujie Han, Alvaro Fuentes, Jongbin Park, Yongchae Jeong, Dong Sun Park

**Affiliations:** 1Department of Electronic Engineering, Jeonbuk National University, Jeonju 54896, Republic of Korea; yao@jbnu.ac.kr (Y.M.); shujiejulie@jbnu.ac.kr (S.H.); afuentes@jbnu.ac.kr (A.F.); jbpark99@jbnu.ac.kr (J.P.); ycjeong@jbnu.ac.kr (Y.J.); 2Core Research Institute of Intelligent Robots, Jeonbuk National University, Jeonju 54896, Republic of Korea; 3Department of Computer Engineering, Mokpo National University, Mokpo 58554, Republic of Korea

**Keywords:** cattle’s face recognition, deep learning, open-set recognition, animal welfare, precision livestock farming

## Abstract

**Simple Summary:**

Over the years, the identification of individual cattle has assumed a pivotal role in health monitoring, reproduction management, behavioral research, and performance tracking. In this study, we propose a method based on artificial intelligence for identifying known and new (unknown) individual Hanwoo cattle, a native breed of Korea, by utilizing cattle’s face images. To accomplish this, we strategically positioned a network of CCTV cameras within a closed farm, demonstrating the efficacy of non-intrusive sensors in capturing real-world data. Furthermore, we devised open-set techniques to tackle challenges such as varying illumination, overlapping objects, and fluctuations in cattle’s face orientations. Our research method not only demonstrated excellent recognition performance in complex real-world cattle’s datasets, but can also be applied to open-set scenarios, wherein unmarked or new cattle may join the herd. Our proposed method can be readily adapted to identifying various livestock species, offering real-time individual recognition, which yields valuable insights for farm management. This deep learning approach amplifies the efficiency of farm operations, thus playing a pivotal role in advancing the agriculture industry as a whole.

**Abstract:**

Accurate identification of individual cattle is of paramount importance in precision livestock farming, enabling the monitoring of cattle behavior, disease prevention, and enhanced animal welfare. Unlike human faces, the faces of most Hanwoo cattle, a native breed of Korea, exhibit significant similarities and have the same body color, posing a substantial challenge in accurately distinguishing between individual cattle. In this study, we sought to extend the closed-set scope (only including identifying known individuals) to a more-adaptable open-set recognition scenario (identifying both known and unknown individuals) termed Cattle’s Face Open-Set Recognition (CFOSR). By integrating open-set techniques to enhance the closed-set accuracy, the proposed method simultaneously addresses the open-set scenario. In CFOSR, the objective is to develop a trained model capable of accurately identifying known individuals, while effectively handling unknown or novel individuals, even in cases where the model has been trained solely on known individuals. To address this challenge, we propose a novel approach that integrates Adversarial Reciprocal Points Learning (ARPL), a state-of-the-art open-set recognition method, with the effectiveness of Additive Margin Softmax loss (AM-Softmax). ARPL was leveraged to mitigate the overlap between spaces of known and unknown or unregistered cattle. At the same time, AM-Softmax was chosen over the conventional Cross-Entropy loss (CE) to classify known individuals. The empirical results obtained from a real-world dataset demonstrated the effectiveness of the ARPL and AM-Softmax techniques in achieving both intra-class compactness and inter-class separability. Notably, the results of the open-set recognition and closed-set recognition validated the superior performance of our proposed method compared to existing algorithms. To be more precise, our method achieved an AUROC of 91.84 and an OSCR of 87.85 in the context of open-set recognition on a complex dataset. Simultaneously, it demonstrated an accuracy of 94.46 for closed-set recognition. We believe that our study provides a novel vision to improve the classification accuracy of the closed set. Simultaneously, it holds the potential to significantly contribute to herd monitoring and inventory management, especially in scenarios involving the presence of unknown or novel cattle.

## 1. Introduction

The demand for livestock products, including meat and dairy, is experiencing nearly exponential growth due to the expanding global population and increased affordability of these commodities. To enhance productivity, minimize resource wastage, promote animal welfare, and facilitate sustainable and efficient livestock farming practices, the precision livestock farming approach has been developed (https://geopard.tech/blog/precision-livestock-farming-technologies-benefits-and-risks/ (accessed on 16 November 2023)). In the context of precision livestock farming management, the identification of individual cattle assumes a pivotal role, encompassing tasks such as health monitoring [1], reproduction management [2], behavior research [3,4,5], and individual cattle’s performance tracking [6]. The widely adopted cattle’s identification technology, Radio Frequency Identification (RFID) [7,8], faces various challenges in practical use, including a limited recognition area, instances of collisions, and the potential for tag duplication and loss. Consequently, the labor costs associated with the application of RFID technology are notably high. As a result, the use of RFID is gradually revealing inefficiencies and cost-related drawbacks. In recent times, the advent of deep-learning-based approaches for cattle identification tasks [9,10,11] has led to notable advancements, driven by their low-cost, non-invasive, and stress-free nature, and efficacy.

Nonetheless, the application of deep-learning-based methods for automated cattle’s face recognition encounters numerous challenges in real-world farm settings, such as issues related to varying illumination, weather conditions, overlapping objects, and fluctuations in cattle’s face orientations. While collecting data during cattle’s feeding, multiple cattle often share the same feeding trough, resulting in overlapping cattle’s faces. Additionally, as cattle move their heads while eating, only partial images of their faces are captured. Moreover, cattle assume various postures and orientations during feeding, leading to significant variations in the angles. Furthermore, since the data are obtained during three distinct feeding times, the same cattle may feed at different troughs, causing variations in the distances between the cattle and the cameras. As a result, the collected data may also feature differences in their characteristics, including variations in head size. These factors present substantial challenges for accurately identifying cattle’s faces. Some examples of the challenges addressed in this study can be illustrated in the dataset collection subsection. Furthermore, these cattle recognition tasks primarily function under the closed-set assumption, which typically implies that the source (training) and target (testing) datasets share the same classes (known classes). This closed-set assumption, while suitable for some scenarios, encounters limitations in real-world farm applications, particularly concerning the emergence and re-identification of new cattle (unknown) within the herd.

To tackle these challenges and overcome the limitations of the closed-set assumption, we embraced a more-adaptable open-set [12] perspective for recognizing known and unknown individual cattle, relying on highly distinguishable cattle’s facial features, termed Cattle’s Face Open-Set Recognition (CFOSR). At the same time, we enhanced the performance of the closed-set classification task by integrating open-set techniques to obtain a good classifier. In Open-Set Recognition (OSR), the training phase involved known individuals, while the test phase accommodated the presence of new (unknown) individuals. Specifically, the model, having been trained solely on known individuals, was expected to accurately identify recognized individual cattle or effectively distinguish previously unseen individuals during the testing stage. As an instance, consider the scenario where a new individual is introduced to the barn. In contrast, in the context of closed-set identification, an unknown individual might erroneously be categorized as one of the known individuals, thereby impeding the accurate monitoring and tracking of individual cattle within real-world farm settings. For instance, this issue becomes particularly pronounced when unidentified cattle carry infectious diseases; the failure to promptly recognize them and institute appropriate measures can potentially inflict significant harm on the entire cattle herd.

A prominent challenge in OSR pertains to the absence of unknown individuals during the training phase, thereby confining the model’s learning solely to known individuals’ information [13]. Moreover, the complexity of CFOSR surpasses that of conventional OSR in the realm of computer vision, predominantly due to the subtle disparities observed among most cattle’s faces. In contrast, prevalent computer vision datasets utilized to evaluate OSR tasks [12,14,15], such as TinyImageNet and CIFAR10, encompass semantically distinct known and unknown classes, such as cats and dogs. In the context of CFOSR, where Hanwoo cattle’s faces have a similar appearance, to obtain a good open-set classifier, a prevailing approach involves cultivating a more-compressed feature space for known individuals, thereby affording greater expansiveness to unknown individuals [16,17]. The whole architecture is shown in Figure 1.

To address the challenges in CFOSR, on one front, we opted for a straightforward, yet impactful loss function known as the Additive Margin Softmax loss (AM-Softmax) [18]. This choice amplifies the separability of distinct individual features, while concurrently compacting the distance between the features of the same individual. This, in turn, facilitates the provision of additional feature spaces for accommodating unknown individuals. On the other hand, we harnessed the potential of the distance-based Adversarial Reciprocal Point Learning (ARPL) loss to curtail the overlap between the known and unknown distributions. Specifically, the reciprocal point for each known class was derived within an extra-class space, followed by the imposition of an adversarial margin constraint, which confined the extent of the latent open space established by these reciprocal points [17].

Prior investigations [15] posited that a good closed-set classifier can offer valuable support for open-set recognition tasks. Within this research, we harnessed transfer learning to acquire an adept closed-set classifier, a strategic move that notably enhanced the performance in the subsequent tasks [19,20]. We opted for a Vision Transformer (ViT) model, pretrained on the ImageNet21K dataset, instead of the more-conventionally employed ImageNet1K dataset. Moreover, we harnessed a ViT model that was pretrained on the plant-relevant dataset PlantCLEF2022 [21]. Interestingly, we observed that the datasets focused on plants also contributed to enhancing the accuracy of the cattle’s face recognition. Additionally, we employed transfer learning with a pretrained ResNet50 model sourced from the ImageNet1K dataset.

The remainder of this paper is organized as follows: In Section 2, we formally define the cattle’s face open-set recognition, introduce our dataset, and provide detailed insights into the proposed method. Section 3 presents the implementation details and experimental results, showcasing the performance of our model and highlighting the significance of our findings. In Section 4, we outline the limitations and highlight key contributions. Finally, in Section 5, we conclude the paper by summarizing the key techniques and suggesting avenues for future research.

## 2. Materials and Methods

### 2.1. Problem Definition

In this research, our goal was to develop a robust classifier for Cattle’s Face Open-Set Recognition (CFOSR) using a real-world farm dataset. This subsection is devoted to formally defining the open-set recognition, specifically CFOSR. Let $D_{train}={\left\{\left(x_{i},y_{i}\right)\right\}}_{i=1}^{N}\subset X_{train}\times Y_{train}$ denote the training dataset, where $X_{train}$ represents the input image space and $Y_{train}$ signifies the corresponding label space. Here, *N* corresponds to the total inputs in the training dataset. Similarly, let $D_{test}={\left\{\left(x_{i},y_{i}\right)\right\}}_{i=1}^{M}\subset X_{test}\times Y_{test}$ characterize the test dataset, with *M* denoting its overall inputs. Operating under the Closed-Set Assumption (CSA), both the training and test datasets share a common label space, $Y_{test}=Y_{train}$. However, in real-world testing scenarios, novel individual cattle may emerge, a situation that poses a risk when current methods classify the new cattle among the known cattle in $X_{train}$. Consequently, there arises a desire to extend the closed-set paradigm to embrace the open-set realm.

Mathematically, the test dataset in CFOSR is expressed as $Y_{test}=Y_{train}+Y_{unknown}$, where $Y_{unknown}\neq ⌀$ pertains to the domain of *unknown* or new individual cattle. A well-trained model is mandated to adeptly categorize a testing image, assigning it to either the known individuals in $Y_{train}$ or the unknown individual $Y_{unknown}$.

### 2.2. Dataset Collection and Preprocessing

Dataset collection: This study solely utilized video data. No physical experiments or intrusive devices that could disrupt the animals’ normal conditions were employed in our research. The dataset comprises video recordings obtained from the “Baksagol” private Hanwoo cattle farm located in Imsil, South Korea. This farm is situated in a temperate climate with distinct seasons, experiencing cold, dry winters and hot, humid summers in South Korea. Spring and autumn are relatively brief, offering mild and generally pleasant temperatures. The animal housing facility was designed with semi-open compartments, allowing for external air ventilation. Additionally, each compartment is equipped with indoor ventilators. The floor is covered with sawdust on a basic concrete foundation, while the ceiling consists of opaque Styrofoam steel sheets in some areas and transparent polycarbonate in others.

The experimental barn had a size of 30 × 12 m and housed 21 cattle, ranging in age from 1 to 7 y, including 3 calves. To capture continuous video data for face recognition, three Hikvision (HIK) surveillance camera devices with 4 K resolution (3840 × 2160) were installed in the barn facing the longest area of the barn with a clear view of the animal faces. Figure 2 illustrates the corresponding camera setup. It is important to highlight that we deployed these three cameras at various locations to capture the data during three distinct feeding periods: morning, noon, and night, as illustrated in Figure 3. To standardize the data, we extracted image frames from the three video cameras at a rate of 15 frames per second (fps). We employed the upper left and lower right coordinates to effectively label the cattle’s face data, distinctly indicating the face’s position, as depicted in the lowermost row of Figure 3. Furthermore, we tracked and annotated each of the cattle based on video data to determine their exact location.

Preprocessing: For the collection of cattle’s face datasets, precise annotations of the bounding boxes were applied to accurately pinpoint the cattle’s faces within the original images. Subsequently, the face images were obtained through cropping based on these annotations. It is important to note that this aspect of the work was not our primary focus; we extend our gratitude to our research collaborator for providing the cattle-to-face image datasets. Our dataset encompasses a span of three days, during which we captured face images across three distinct feeding instances. To enhance the model training, we merged the images from all three feeding times for a given day, as illustrated in the parity examples displayed in Figure 4. As shown in the figure, it is clear that each of the cattle’s faces exhibited multiple angles and orientations, and some images were taken in foggy conditions, which added complexity to the dataset. Notably, the figure also illustrates that distinct cattle often share remarkably similar facial features, significantly augmenting the challenge of accurate cattle identification. In our experimental setup, images from one day were allocated as the training dataset, while the other two days’ images were the testing datasets (Testing Dataset 1 and Testing Dataset 2). In Testing Dataset 2, there are relatively few instances of overlapping images of two cattle and a few images containing only a tiny portion of the cattle’s faces. Further details regarding the number of images corresponding to each of the individual cattle in both the training and testing datasets can be found in Table 1.

### 2.3. Proposed Method

To achieve a well-trained open-set classifier within the CFOSR context, we devised a unified algorithm by incorporating two strategies into a method for open-set recognition of cattle faces.

#### 2.3.1. Cattle’s Face Open-Set Recognition

In this section, our aim was to introduce a baseline approach for CFOSR. We leveraged Adversarial Reciprocal Points Learning (ARPL) [17], one of the state-of-the-art techniques in the domain of open-set recognition. The architectural layout, as depicted in Figure 1, encompasses the learning of *K* known reciprocal points during the training phase, where *K* signifies the count of known individuals. At its core, each input image *x* traverses through the feature extractor *f* to yield feature representations, denoted as f(x). These learned features for the known entities are strategically positioned to exhibit a notable separation from their corresponding reciprocal points. During the evaluation phase, a given sample is allocated to either the known or unknown category based on the calculated distance between its features and the reciprocal points. To elaborate, if the distance to all reciprocal points falls below a predefined threshold τ, the model designates the sample as an unknown entity. Conversely, the model assigns samples to specific known classes based on the maximum distance between the feature and all reciprocal points. The ARPL loss is combined with the cross-entropy loss and Adversarial Margin Constraint (AMC) loss. Notably, given sample *x* and reciprocal point $P_{k}$, their distance $d(f\left(x\right),P_{k})$ is calculated by combining the Euclidean distance and dot product when using the cross-entropy loss, while the AMC loss only uses the Euclidean distance $d_{e}\left(f\left(x\right),P_{k}\right)$ to mitigate the overlap between the distributions of known and unknown individuals, which is given by:(1)$$L_{AMC}=max\left(d_{e}\left(f\left(x\right),P_{k}\right)-R,0\right),$$
where *R* is a learnable margin.

#### 2.3.2. Additive Margin Softmax to Enhance Compactness within the Known Feature Space

Our first strategy was to employ the additive margin softmax loss rather than the cross-entropy loss, to obtain a compact feature space. Different from closed-set classification, OSR models are tasked with generating an unknown score to indicate the likelihood of an input sample belonging to the unknown category. Fine-tuning a threshold can make a decision based on this unknown score. In this case, the advantage of a compact feature space for known classes becomes apparent since it allocates more space for unknown ones. Inspired by this idea, the Additive Margin Softmax loss (AM-Softmax) [18] function was employed, which can be formalized as
(2)$$L_{AM}=-\frac{1}{n}\sum_{i=1}^{n}\log\frac{e^{s\cdot (W_{\hat{c}}^{T}f_{i}-m)}}{e^{s\cdot (W_{\hat{c}}^{T}f_{i}-m)}+\sum_{c=1,c\neq \hat{c}}^{C}e^{s\cdot W_{c}^{T}f_{i}}},$$
where *n* is the total number of samples, *C* is the total number of known classes, and $\hat{c}$ is the correct class. The hyperparameter margin *m* is only used for the correct class, and *s* is a scaling hyperparameter. In the equation, f denotes extracted features for an input sample, and W is the weights in the classifier layer. Notably, the feature vector $f_{i}$ and weight vector W need to be normalized. The AM-Softmax with the margin *m* added to the decision boundary increases the separability of the classes and makes the distance between the same classes more compact in CFOSR, as shown in Figure 5.

#### 2.3.3. Transfer Learning

Our second strategy was utilizing transfer learning to boost the classification performance. Transfer learning aims to leverage the knowledge learned from source tasks in different domains to adapt to target tasks, so it does not need to learn from scratch with large amounts of data [22,23,24]. Benefiting from transfer learning, the model can attain enhanced performance within a relatively short training period. In this study, we leveraged a large ViT model pretrained on the ImageNet21K dataset. This dataset contains a greater diversity of classes and images compared to the ImageNet1K dataset. Better performance in the target task is often observed when the source dataset comprises a wider array of classes and substantial images [25]. Moreover, we opted for a ViT model that was pretrained on the plant-relevant dataset PlantCLEF2022 [21]. Interestingly, we observed that the plant-relevant datasets also contributed to enhancing the performance of the cattle’s face identification. In the case of the CNN-based model, utilizing a pretrained model on ImageNet1k led to improved accuracy. As such, we employed the ResNet50 model, initially pretrained on the ImageNet1k dataset, and subsequently, fine-tuned it using the cattle’s face dataset.

## 3. Results

### 3.1. Evaluation Metrics

AUROC: CFOSR encompasses two core tasks: distinguishing unknown individuals from known ones and accurately classifying known individuals. Due to the unique nature of these tasks, the conventional metrics utilized in generic image classification may not be suitable for assessing CFOSR’s performance [26]. An initial challenge in evaluating CFOSR stems from the fact that varying thresholds for identifying unknown from known individuals can yield disparate performance outcomes. An effective metric for addressing this concern is the Area Under the Receiver Operating Characteristic (AUROC) curve [27], which remains robust to threshold variations. The computation of the AUROC involves consolidating all known individuals into a super-known individual class, effectively transforming the task into binary classification alongside a super-unknown individual class encompassing the unknown individuals. Within this framework, the True Positive Rate (TPR) and False Positive Rate (FPR) are calculable for the super-known individual. As implied by the term AUROC, it represents the area under the curve formed by $TPR_{s}$ and $FPR_{s}$. A higher AUROC value typically indicates a superior model performance.

OSCR [28]: This assessment methodology simultaneously addresses both CFOSR tasks, with the objective of yielding a singular value for evaluating the trained models. The test datasets are partitioned into distinct known individuals $D_{known}$ and unknown individuals $D_{unknown}$. Regarding samples originating from $D_{known}$, the Correct Classification Rate (CCR) signifies the proportion of samples for which the unknown scores fall below a designated threshold *s*, while the learned classifier maintains accurate classification. Conversely, for samples stemming from $D_{unknown}$, the False Positive Rate (FPR) corresponds to the fraction of samples with unknown scores below *s*. Similar to the AUROC, the OSCR metric denotes the area under the curve traced by $CCR_{s}$ and $FPR_{s}$.

CSA: While the AUROC and OSCR provide valuable insights into assessing known and unknown performance, they lack the capacity to evaluate the model’s precision in classifying known classes accurately. To address this aspect, the concept of Close-Set Accuracy (CSA) [15,27] proves useful. CSA can be employed conjointly with the former metrics, ensuring that a proficient open-set classifier maintains its efficacy in closed-set scenarios. By exclusively considering known classes for model training, CSA equates to the conventional accuracy metric. Hence, the combined utilization of the AUROC, OSCR, and CSA offers a comprehensive framework for evaluating models while simultaneously considering both tasks.

### 3.2. Implementation Details

As previously noted, our training dataset comprises images from three feeding times for 5 November 2021, while the testing dataset encompasses images from November 6 and 9 November 2021. From the entire pool, ten known individuals were chosen randomly, leaving seven individuals as unknown entities. For data augmentation, we employed RandAugment [29] with parameter $AUG_{m}$ set at 30 and parameter $AUG_{n}$ set at 2. The images were subsequently resized to dimensions of 224. Regarding the AM-Softmax function, initial experimentation guided the selection of parameters *s* and *m*, which were set at 10.0 and 0.5, respectively.

In the training process, a batch size of 32 was utilized, along with 16 workers, leveraging a single NVIDIA (Santa Clara, CA, USA) RTX 3090 GPU equipped with 24 GB of memory. The network underwent training for 100 epochs, guided by a learning rate of 0.0001. Following the conventional experimental setup for OSR, each model was subjected to training across five distinct random splits of known and unknown entities within the dataset. The mean and variance of the model’s performance across these splits are subsequently reported.

We trained our model end-to-end by incorporating both the $L_{AMC}$ and $L_{AM}$ loss functions. The $L_{AMC}$ loss function was employed to mitigate the overlap between the known class space and the remaining space (unknown class space), while $L_{AM}$ was utilized for classifying known individuals.

### 3.3. Compared Methods

In order to validate the effectiveness of our approach, we conducted comparisons with several methods. The specific introductions of the comparative methods are outlined as follows:RN50. A ResNet50 model was trained from scratch with the cattle’s face datasets.RN50-IN1k. A ResNet50 model was pretrained with the ImageNet-1k (IN1k) dataset in a supervised way and, then, fine-tuned in the cattle’s face datasets.ViT-L. A large ViT [30] model was trained from scratch with the cattle’s face datasets.ViT-MAE. A large ViT model was pretrained with the IN1k dataset in a self-supervised way [31].ViT-PlantCLEF. We pretrained the large ViT model from the MAE with the PlantCLEF2022 dataset once again in a supervised way [32] and, then, fine-tuned it with the cattle’s face datasets.ViT-IN21k. A large ViT model was pretrained with the ImageNet-21k (IN21k) dataset in a supervised way.ViT-IN21k-AM (ours). A large ViT model was pretrained with the ImageNet-21k (IN21k) dataset in a supervised way. Furthermore, the AM-Softmax was leveraged instead of the cross-entropy loss.RN50-IN1k-AM (ours). A ResNet50 model was pretrained with the ImageNet-1k (IN1k) dataset in a supervised way. Furthermore, the AM-Softmax was leveraged instead of the cross-entropy loss.

### 3.4. Results for Open-Set Scenario

#### 3.4.1. Main Results

As one of our main objectives was to achieve cattle’s face open-set recognition in a real-world dataset, we first compared our method to other strategies. Table 2 and Table 3 denote the main results of different comparison methods with 10 and 7 cattle as known classes, respectively.

As shown in Table 2, the experimental results showed that our methods significantly outperformed other methods. In Testing Dataset 1 (TD1), our proposed method, ViT-IN21k-AM, attained a performance of 94.49 CSA, 91.84 AUROC, and 87.85 OSCR. This reflects a significant improvement of 16.78 CSA, 13.47 AUROC, and 21.93 OSCR in comparison to the ViT-MAE method. Furthermore, our CNN-based approach, RN50-IN1k-AM, also exhibited favorable results with 93.61 CSA, 90.84 AUROC, and 87.22 OSCR. Notably, the AM-Softmax function played a crucial role in promoting CFOSR. Specifically, the AM-Softmax significantly enhanced the CSA, AUROC, and OSCR by 4.45, 5.73, and 8.02, respectively, in the CNN-based method. More detail will be shown in the ablation study section.

In the evaluation of Testing Dataset 2 (TD2), the effectiveness of our proposed approach, ViT-IN21k-AM, was evident as it attained a performance of 98.74 CSA, 95.12 AUROC, and 94.33 OSCR, benefiting from the refined data. Meanwhile, the RN50-IN1k-AM method yielded results of 95.98 CSA, 90.65 AUROC, and 88.95 OSCR. Figure 6 shows the known and unknown density distribution with different methods. In contrast to the ViT-MAE approach (depicted in Figure 6d), our proposed method, ViT-IN21k-AM (illustrated in Figure 6a), effectively distinguished between known and unknown individuals. Considering the effect of the AM-Softmax, both the ViT-IN21k-AM (Figure 6a) and RN50-IN1k-AM (Figure 6e) methods exhibited enhanced discrimination between known and unknown individuals, outperforming the ViT-IN21K (Figure 6b) and RN50-IN1k (Figure 6f) methods.

In terms of the influence of transfer learning, both the CNN-based and ViT-based methods notably surpassed the results obtained by training the ResNet50 and ViT models from scratch in the context of CFOSR. A phenomenon worthy of attention in Table 2 is that, even when pretrained on plant-related datasets (PlantCLEF2022), the ViT model continued to achieve superior performance when fine-tuned on the cattle’s face dataset. The same phenomenon can be found in Figure 6c,d. In this context, it was plausible that the features related to texture and color could be transferred to the cattle’s face dataset [33]. When compared to being pretrained on ImageNet1k, the ViT model pretrained on the larger dataset ImageNet21k demonstrated significantly improved performance. To be specific, the ViT-IN21k method exhibited a notable improvement of 15.86 CSA, 12.87 AUROC, and 21.43 OSCR compared to the ViT-MAE method on Testing Dataset 1 (TD1). The preliminary experimental results indicated that fine-tuning played an important role in enhancing the performance of open-set cattle’s face recognition.

Table 3 illustrates a similar trend to Table 2. Furthermore, our proposed methods exhibited comparable performance when dealing with a reduced number of known classes. The influence of openness on CFOSR will be detailed in the openness section.

#### 3.4.2. Ablation Study and Visualization in CFOSR

For the purposes of evaluating our method, this section primarily focuses on analyzing the influence of the AM-Softmax function on open-set scenarios based on Testing Dataset 1 (TD1). Table 4 displays the results of four methods employing the cross-entropy loss or AM-Softmax. Thanks to the utilization of the AM-Softmax, we attained both intra-class compactness and inter-class separability, resulting in improved identification accuracy and enhanced performance in open-set scenarios. The RN50-IN1k method, incorporating the AM-Softmax function, attained results of 93.61 CSA, 90.84 AUROC, and 87.22 OSCR. This demonstrated a notable improvement of 4.45 in the CSA, 5.73 in the AUROC, and 8.02 in the OSCR compared to using the cross-entropy loss. The AM-Softmax also played a crucial role in advancing the ViT-based model. Figure 7 illustrates the observed improvement trend in the AUROC and OSCR subsequent to the implementation of the AM-Softmax.

To analyze the influence of the AM-Softmax in more detail, we attained the confusion matrix of known individuals from the ViT-IN21k and RN50-IN1k methods, as shown in Figure 8. The application of the AM-Softmax enhanced the accuracy of individual cattle identification for both the ViT-IN21k and RN50-IN1k methods. Especially for the Cattle 7, with the utilization of the ViT-IN21k-AM and RN50-IN1k-AM methodologies, the individual cattle were precisely identified with probabilities of 98% and 92%, respectively. Another notable phenomenon was that, compared to the RN50-IN1k-AM method, the ViT-IN21k-AM method could correctly identify most individual cattle with high probabilities. Especially for Cattle 1, the ViT-IN21k-AM method recognized this individual with a probability of 90%, while RN50-IN1k-AM only identified it with a probability of 72%.

Furthermore, based on the RN50-IN1k-AM method, we conducted an analysis of the effect of the margin *m* on the AM-Softmax function. As depicted in Table 5, the parameter *m* had limited influence on model performance. The model achieved relatively good performance when the *m* value was set to 0.5, leading us to select 0.5 as the final value for *m*.

#### 3.4.3. Openness

To analyze the impact of openness on CFOSR, we introduced openness based on the ratio of the numbers of individuals in the training and test sets [13].
(3)$$openness=1-\sqrt{\frac{k_{train}}{k_{test}}},$$
where $k_{train}$ and $k_{test}$ are the number of individuals in the training dataset and the test dataset, respectively.

For a more-comprehensive examination of the impact of known ones on CFOSR, we introduced a novel sampling method to regulate the count of known and unknown individuals. Analogous to our previous experiments, we subjected the “known and unknown” individual allocations to random division in five distinct trials. In addition, we incorporated a distinctive approach by maintaining a consistent number of unknown individuals across all trials while progressively reducing the number of known individuals. Specifically, for the cattle’s face dataset, we upheld seven unknown individuals as a constant, while varying the known individuals’ count from ten down to one. This configuration ensured that a higher degree of openness was achieved with fewer known individuals, underscoring a pronounced relationship between known and unknown individuals.

Figure 9 depicted the impact of different degrees of openness on performance. As the count of known individuals diminished, both the AUROC and OSCR values exhibited a corresponding decline. Notably, when openness reached 64.64%, with one of the cattle was included in the training dataset, a rapid performance deterioration was observed. This outcome was attributed to the heightened challenge of discerning unknown entities in the absence of substantial known information.

Conversely, in scenarios characterized by a reduced number of known individuals during training, the classifier adeptly achieved precise classifications of these known entities. Consequently, the CSA value experienced an upward trajectory. This phenomenon was indicative of the classifier’s enhanced ability to accurately categorize known individuals under the circumstances of limited known individuals in the training data.

### 3.5. A Well-Trained Open-Set Classifier Boosts Closed-Set Recognition

The integration of open-set techniques such as the ARPL loss and AM-Softmax in CFOSR served to enhance the intra-class compactness and inter-class separability. This strengthened the model’s capacity to precisely classify classes within closed-set cattle’s face recognition. In this section, our primary focus is on comparing our methods with the frequently employed cross-entropy loss in closed-set cattle’s face recognition. The comparative experiments were conducted based on the more-challenging Testing Dataset 1 (TD1). We mainly used the Accuracy (Acc) and F1-score as the metrics to evaluate the models.

As depicted in Table 6, when employing the classifier derived from the trained CFOSR model, we attained both better performance on the CNN-based and ViT-based methods. Specifically, we achieved an accuracy of 93.43 and an F1-score of 93.49 with the CNN-based method (RN50-IN1k-AM). This signifies a notable improvement of 4.63 in accuracy and 4.91 in the F1-score compared to using the cross-entropy loss. Furthermore, our ViT-IN21k-AM method achieved an accuracy of 94.46 and an F1-score of 94.53, surpassing the ViT-IN21k method by 1.84 in accuracy and 2.03 in the F1-score.

Examining the t-SNE distribution shown in Figure 10, it becomes obvious that the utilization of the ARPL loss and AM-Softmax led to a greater distinction between the feature spaces of different individuals and a more-condensed feature space for the same individuals. This effect was particularly pronounced in the case of RN50-IN1k-AM, where clear inter-class separability and intra-class compactness were visible. Benefiting from the open-set techniques, the ViT-IN21k-AM and RN50-IN1k methods can accurately identify individual cattle with high probabilities, as shown in the confusion matrix in Figure 11. Utilizing cross-entropy loss, the RN50-IN1k-CE approach was able to identify Cattle 1 at a mere 35% accuracy (Figure 11d). In contrast, the RN50-IN1k-AM method achieved a significantly higher accuracy of 90% in identifying Cattle 1. Both Cattle 16 and Cattle 11 exhibited higher identification accuracy through the RN50-IN1k-AM method compared to the RN50-IN1k-CE approach. Meanwhile, compared to the ViT-IN21k-CE method, the ViT-IN21k-AM method was able to identify both Cattle 12 and Cattle 1 with higher accuracy.

### 3.6. Computational Complexity

The quantity of parameters and the computational complexity are showcased in Table 7 for two basic architectures, namely the Convolutional Neural Network (CNN) and Vision Transformer (ViT). It is worth noting that the training duration is likewise presented in the table; however, we emphasize that this training duration was influenced by factors beyond just the models themselves. Elements such as the datasets, hardware configurations, and training parameters (e.g., the number of data reading workers in PyTorch) collectively contributed to this training time variation. Additionally, the training time encompassed the evaluation period due to the substantial volume of images within both the training and testing datasets, consequently leading to a relatively extended time frame. From the table, we observe that the ViT-L models required more time, but exhibited a higher level of tolerance. When considering frames per second (fps), the ViT models met the real-time demand, achieving 20 fps.

### 3.7. Limitations

The accessibility of appropriate open-source cattle’s face datasets is still a limitation, prompting us to primarily depend on our proprietary cattle’s face dataset for the method’s validation. Furthermore, leveraging a pretrained model from a substantial animal or animal face dataset can significantly benefit the cattle’s face recognition tasks. However, locating such public pretrained models is challenging.

## 4. Discussion

Accurate identification of individual cattle holds significant importance in farm management, facilitating the monitoring of cattle behavior, disease prevention, and improving animal welfare. This approach allows managers to promptly understand the situation of individual cattle, enabling them to respond promptly to identified issues. Therefore, it can greatly enhance the efficiency, production performance, and health of livestock management, contributing to the sustainability and profitability of the livestock industry.

In contrast to prior research employing wearable devices, the current study introduced a non-invasive approach utilizing image data. This innovative method involved the placement of multiple cameras on a real-world closed farm, capturing data during feeding times, and promoting the use of non-invasive information for individual cattle’s identification. The gathered data were subsequently processed by the proposed deep-learning-based architecture to accurately identify individual cattle and, at the same time, recognize unknown individuals.

The qualitative and quantitative results obtained from both closed-set and open-set scenarios validated the effectiveness of the proposed techniques. Furthermore, employing state-of-the-art classifiers and metrics allowed for comparative analysis, revealing significant potential for further improvements in future research. However, a significant limitation of the current model is that it identified all unknown classes as a single unknown class and could not further differentiate among these unknown classes. Addressing this limitation requires the incorporation of additional techniques to distinguish among unknown classes. For instance, we can apply clustering techniques to classify unknown classes. Additionally, there is room for improvement in recognition accuracy when dealing with more-complex datasets. Moreover, obtaining image annotations from video data has been proven to be a complex and time-consuming task. Therefore, additional research and technology are needed to replicate the proposed framework in a more-versatile system that can operate across multiple farms. Our future research efforts will be dedicated to addressing these challenges.

## 5. Conclusions

In this study, we proposed a method to achieve cattle’s face recognition in the open-set scenario named CFOSR. Meanwhile, from a novel perspective, introducing an effective open-set classifier has the potential to significantly enhance the classification performance in closed-set scenarios. To obtain an effective classifier in the CFOSR context, two strategies were utilized and incorporated with the state-of-the-art OSR method, the APRL. To be more specific, the AM-Softmax was employed to have a compact intra-class feature space that is beneficial to detect the unknown ones. A ViT-based model pretrained in a large-scale dataset, ImageNet21k, was transferred for the downstream tasks, compared to the commonly used small-scale dataset ImageNet1K. Furthermore, we observed that the plant-relevant dataset PlantCLEF2022 also contributed to enhancing the performance of the cattle’s face identification. Our strategies were executed on real-world cattle’s face datasets, and the experimental results validated their effectiveness. More precisely, our method achieved an AUROC of 91.84 and an OSCR of 87.85 in the context of open-set recognition on a complex dataset. Simultaneously, it demonstrated an accuracy of 94.46 for closed-set recognition. Notably, we achieved a 95.12 AUROC and a 94.33 OSCR on a less-challenging dataset. In spite of the decent performance and some basic understanding of CFOSR, we desire to improve our model for real-world applications. We hope our work will contribute to the community, encourage more work, and offer a novel visual approach to enhance closed-set classification accuracy.

## Figures and Tables

**Figure 1 animals-13-03588-f001:**
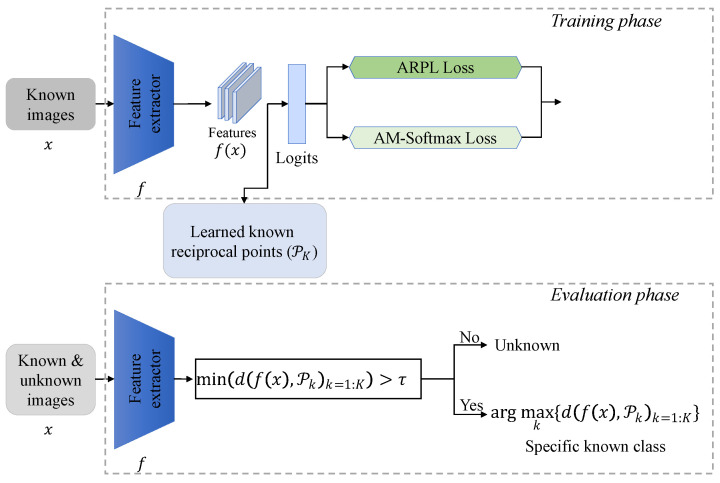
Architecture of our proposed CFOSR method. In the training phase, only known images are fed into the feature extractor. After training the model with three loss functions, the known reciprocal points ($P_{K}$, where *K* is the number of known individuals) are learned. In the evaluation phase, unknown ones are identified based on the distance between the features and the known reciprocal points. *d* is a distance function, and τ is a threshold.

**Figure 2 animals-13-03588-f002:**
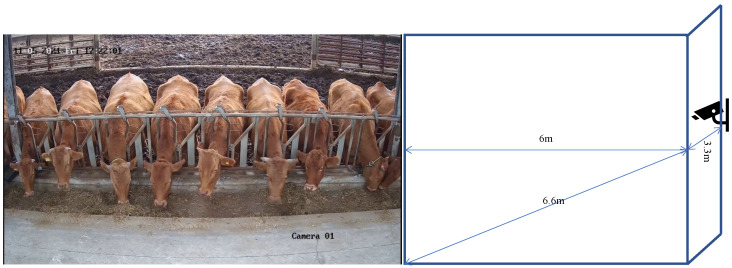
Illustration of camera installation. This figure illustrates the installation diagram of one of the cameras. We positioned a camera on the wall facing the cowshed, placing it at a height of 3.3 m above the ground and at a distance of 6 m from the cowshed.

**Figure 3 animals-13-03588-f003:**
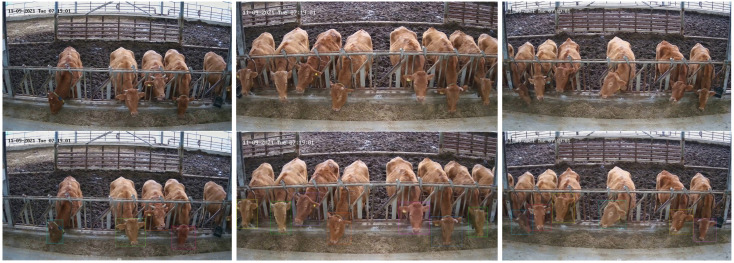
One example of the collected dataset for the morning feeding time. The first row is the images captured by the three cameras, respectively. The second row is the corresponding annotated images with bounding boxes.

**Figure 4 animals-13-03588-f004:**
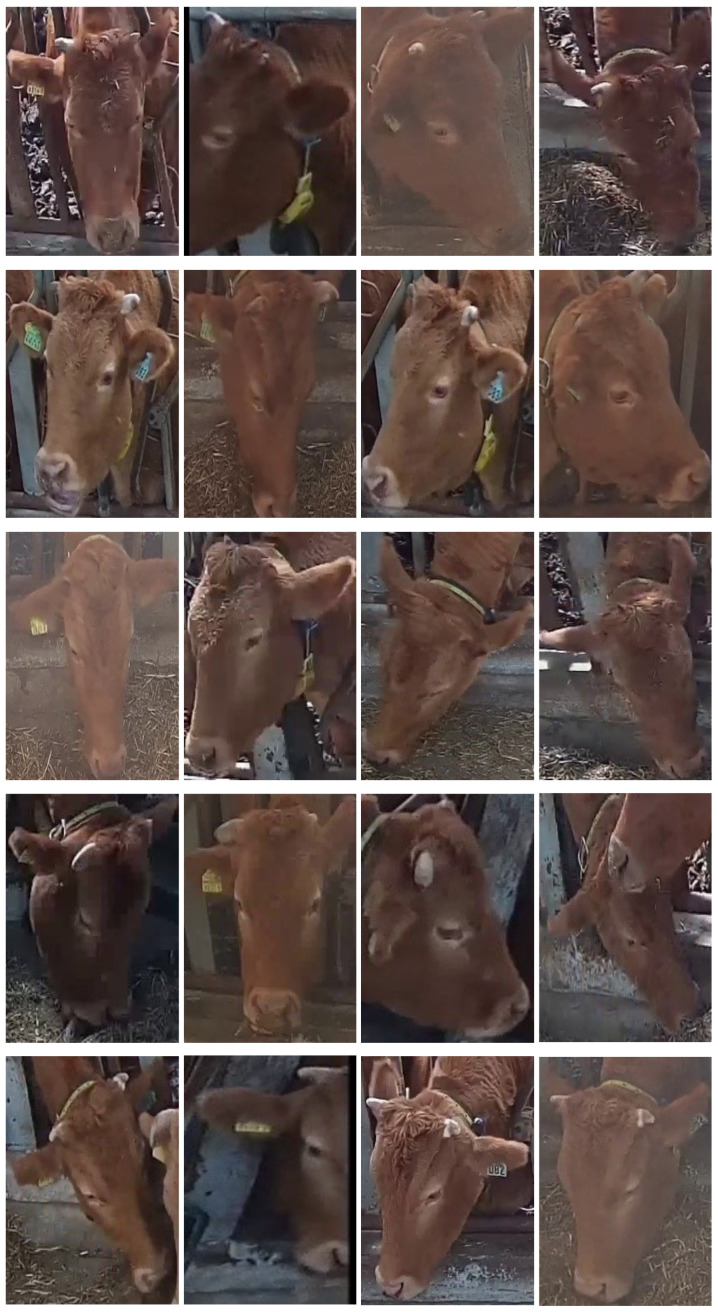
Five instances of cattle are showcased in the training datasets as examples, with each of the cattle encompassing images captured during three distinct feeding instances. Those examples illustrate the challenges faced in this study, including illumination, overlapping cattle’s faces, captured partial cattle’s faces, various postures and orientations, and varying sizes of heads.

**Figure 5 animals-13-03588-f005:**
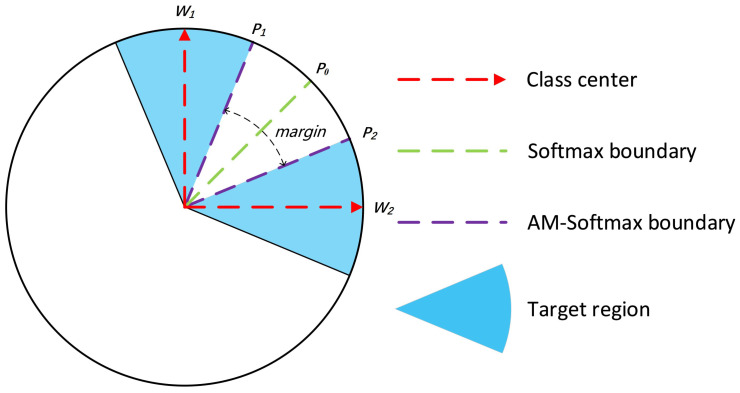
Contrast between decision boundaries: softmax loss (cross-entropy) and AM-Softmax [18]. For the softmax loss, the decision boundary is at P0. For AM-Softmax, the decision boundaries for Class 1 and Class 2 are at P1 and P2, respectively.

**Figure 6 animals-13-03588-f006:**
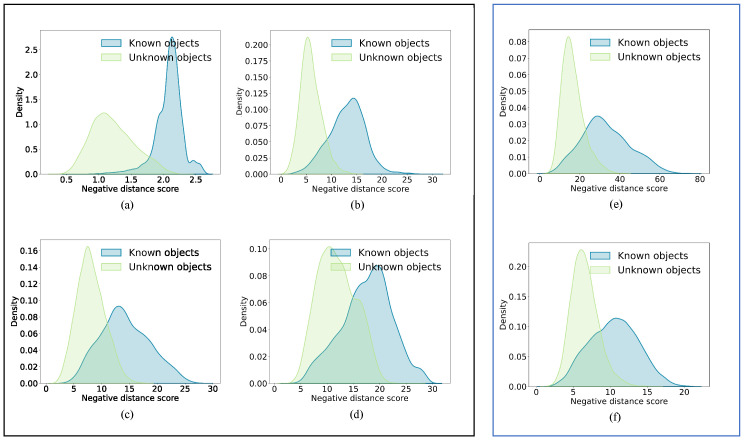
An instance showcasing both known and unknown density distributions in CFOSR. Figures (**a**–**d**) in the black frame indicate ViT-IN21k-AM, ViT-IN21k, ViT-PlantCLEF, and ViT-MAE, respectively. Figures (**e**,**f**) in the blue frame denote RN50-IN1k-AM and RN50-IN1k, respectively. Zoom in to see the details.

**Figure 7 animals-13-03588-f007:**
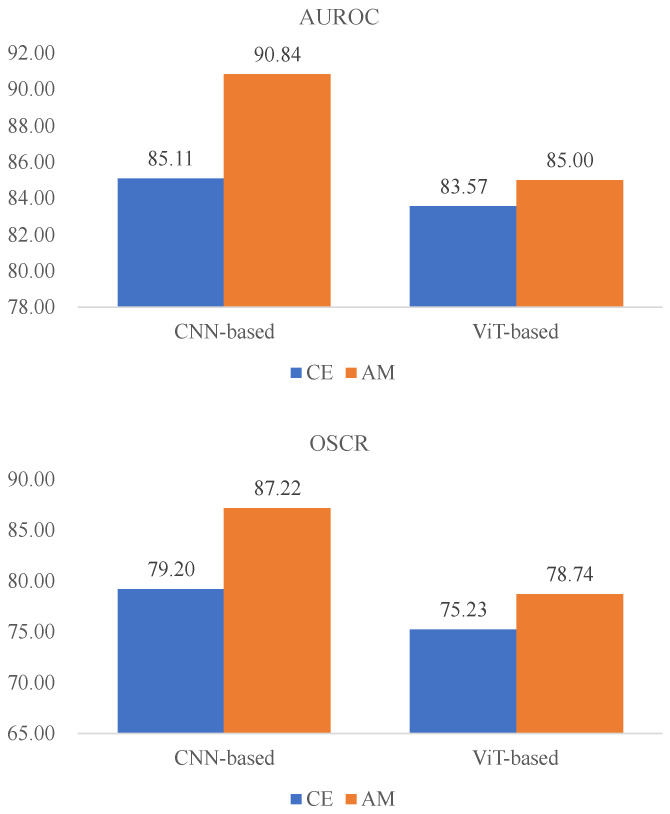
Comparison of AUROC and OSCR results utilizing the cross-entropy loss and AM-Softmax. The ViT-based results represent the average outcomes across ViT-MAE, ViT-PlantCLEF, and ViT-IN21k.

**Figure 8 animals-13-03588-f008:**
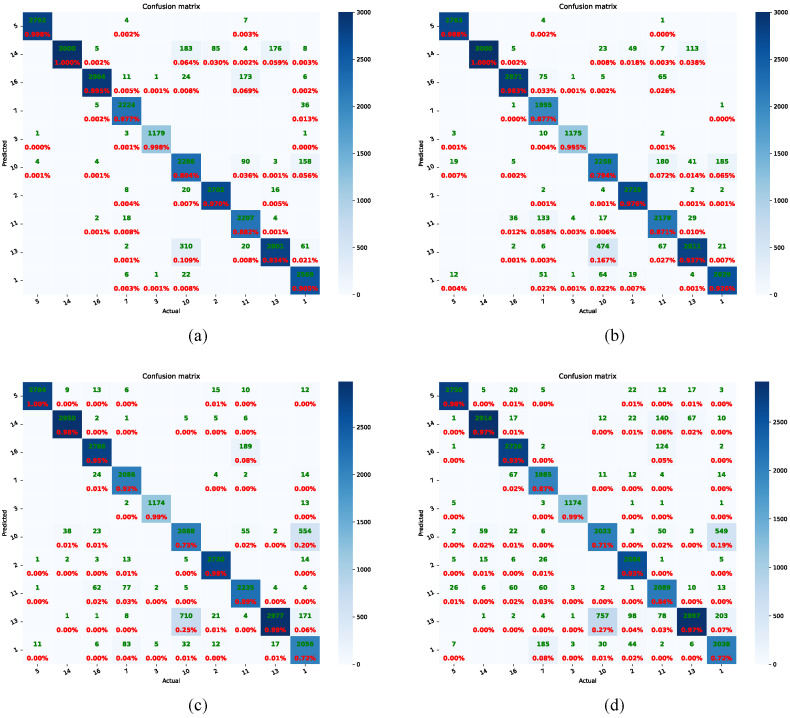
One of the examples of the confusion matrix of known individuals in CFOSR. Figures (**a**–**d**) indicate ViT-IN21k with AM-Softmax (ViT-IN21k-AM), ViT-IN21k with cross-entropy loss (ViT-IN21k-CE), RN50-IN1k with AM-Softmax (RN50-IN1k-AM), and RN50-IN1k with cross-entropy loss (RN50-IN1k-CE), respectively. The green font indicates the number of recognized instances of cattle, while the red font indicates the recognition percentage. Zoom in to see the details.

**Figure 9 animals-13-03588-f009:**
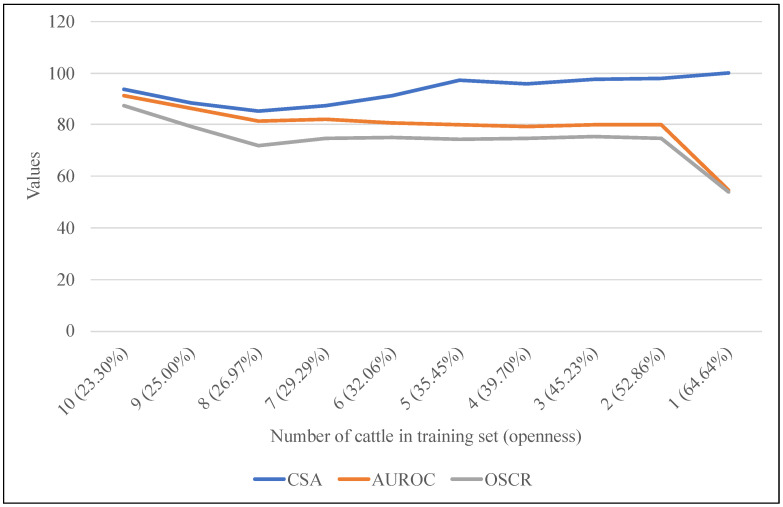
Model performance based on varying numbers of known cattle (openness) on the cattle’s face dataset.

**Figure 10 animals-13-03588-f010:**
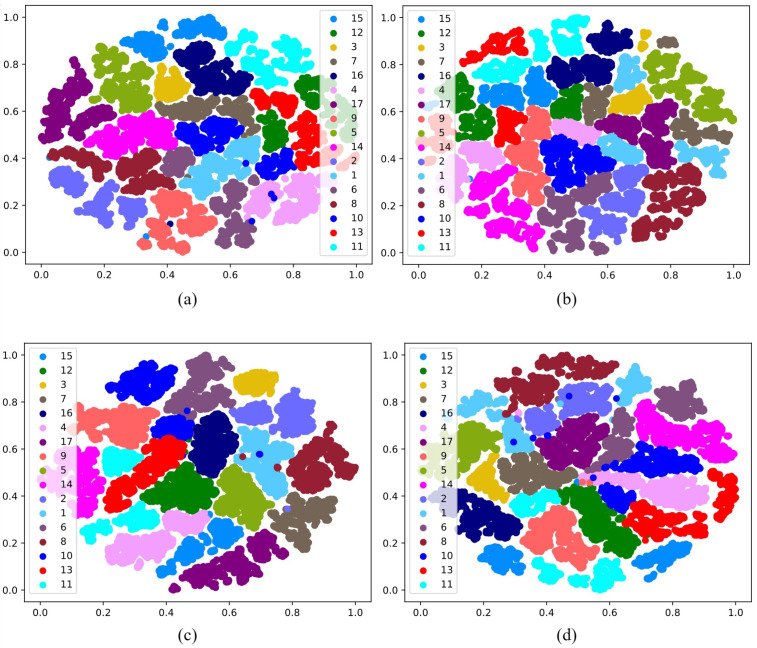
Visualization of t-SNE on cattle’s face dataset with all 17 individual cattle. Figure (**a**–**d**) ViT-IN21k-AM with open-set techniques, ViT-IN21k with cross-entropy loss, RN50-IN1k-AM with open-set techniques, and RN50-IN1k with cross-entropy loss, respectively. Zoom in to see the details.

**Figure 11 animals-13-03588-f011:**
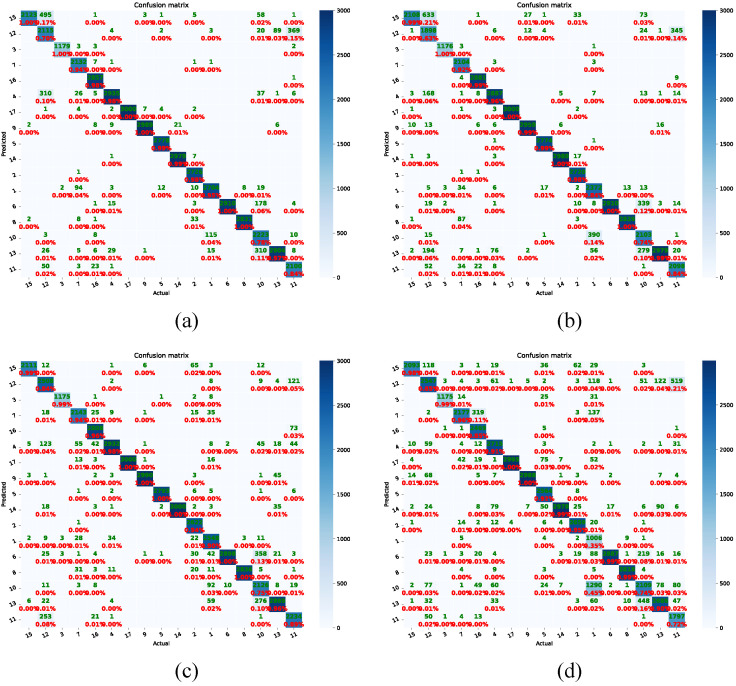
Examples of confusion matrix of all 17 individual cattle. Figures (**a**–**d**) indicate ViT-IN21k-AM with open-set techniques, ViT-IN21k with cross-entropy loss, RN50-IN1k-AM with open-set techniques, and RN50-IN1k with cross-entropy loss, respectively. The green font indicates the number of recognized instances of cattle, while the red font indicates the recognition percentage. Zoom in to see the details.

**Table 1 animals-13-03588-t001:** The number of images of each of the cattle in the training datasets and the testing datasets. Compared to Testing Dataset 1, there are relatively few instances of overlapping images of two cattle and a few images containing only a tiny portion of the cattle’s faces in Testing Dataset 2.

	Training Dataset	Testing Dataset 1	Testing Dataset 2
1	2959	2838	2652
2	2000	2845	2688
3	2670	2501	3000
4	3000	3000	2986
5	3000	3000	3000
6	3000	3000	2656
7	3000	2127	2644
8	3000	2920	2952
9	3000	3000	2645
10	2923	2787	2708
11	3000	1181	1627
12	2948	3000	2929
13	3000	2797	2817
14	2930	2910	2973
15	2990	2276	2457
16	3000	2839	2977
17	2966	3000	2000
total	49,386	46,021	45,711

**Table 2 animals-13-03588-t002:** Main results with different methods. We utilized 10 cattle as known classes and 7 cattle as unknown classes. TD1 and TD2 denote Testing Dataset 1 and Testing Dataset 2, respectively. The boldface represents the best results in the cattle’s face dataset for the specific evaluation metrics.

TD	Method	Training Strategies	CSA	AUROC	OSCR
TD1	RN50	-	30.50 ± 0.04	57.15 ± 0.05	19.94 ± 0.03
	RN50-IN1k	ImageNet1k	89.16 ± 0.03	85.11 ± 0.03	79.20 ± 0.03
	RN50-IN1k-AM (ours)	ImageNet1K	93.61 ± 0.02	90.84 ± 0.03	87.22 ± 0.02
	ViT-L	-	34.42 ± 0.09	55.35 ± 0.05	23.21 ± 0.07
	ViT-MAE	ImageNet1k	77.71 ± 0.03	78.37 ± 0.05	65.92 ± 0.04
	ViT-PlantCLEF	PlantCLEF2022	82.56 ± 0.04	82.10 ± 0.03	72.41 ± 0.02
	ViT-IN21k	ImageNet21k	93.57 ± 0.02	91.24 ± 0.02	87.35 ± 0.01
	ViT-IN21k-AM (ours)	ImageNet21k	**94.49 ± 0.02**	**91.84 ± 0.01**	**87.85 ± 0.02**
TD2	RN50	-	30.89 ± 0.06	51.92 ± 0.04	18.68 ± 0.05
	RN50-IN1k	ImageNet1k	92.16 ± 0.02	86.48 ± 0.02	82.81 ± 0.03
	RN50-IN1k-AM (ours)	ImageNet1K	95.98 ± 0.01	90.65 ± 0.03	88.95 ± 0.03
	ViT-L	-	45.01 ± 0.06	60.43 ± 0.05	32.78 ± 0.05
	ViT-MAE	ImageNet1k	88.48 ± 0.03	80.88 ± 0.04	75.95 ± 0.04
	ViT-PlantCLEF	PlantCLEF2022	89.69 ± 0.03	87.31 ± 0.02	81.35 ± 0.02
	ViT-IN21k	ImageNet21k	97.81 ± 0.01	95.02 ± 0.01	93.81 ± 0.01
	ViT-IN21k-AM (ours)	ImageNet21k	**98.74 ± 0.01**	**95.12 ± 0.02**	**94.33 ± 0.02**

**Table 3 animals-13-03588-t003:** Main results with 7 cattle as known classes and 10 cattle as unknown classes. TD1 and TD2 denote Testing Dataset 1 and Testing Dataset 2, respectively. The boldface represents the best results in the cattle’s face dataset for the specific evaluation metrics.

TD	Method	Training Strategies	CSA	AUROC	OSCR
TD1	RN50-IN1k	ImageNet1k	91.32 ± 0.02	87.26 ± 0.04	83.68 ± 0.04
	RN50-IN1k-AM (ours)	ImageNet1K	**93.99 ± 0.01**	**90.47 ± 0.02**	**87.58 ± 0.03**
	ViT-MAE	ImageNet1k	83.87 ± 0.05	83.19 ± 0.04	73.65 ± 0.04
	ViT-PlantCLEF	PlantCLEF2022	84.50 ± 0.04	84.29 ± 0.04	76.41 ± 0.04
	ViT-IN21k	ImageNet21k	91.62 ± 0.03	89.05 ± 0.02	85.23 ± 0.01
	ViT-IN21k-AM (ours)	ImageNet21k	92.42 ± 0.01	89.42 ± 0.03	85.47 ± 0.02
TD2	RN50-IN1k	ImageNet1k	94.90 ± 0.02	91.38 ± 0.03	88.30 ± 0.04
	RN50-IN1k-AM (ours)	ImageNet1K	97.55 ± 0.01	93.96 ± 0.01	92.57 ± 0.02
	ViT-MAE	ImageNet1k	90.72 ± 0.03	86.80 ± 0.04	81.64 ± 0.05
	ViT-PlantCLEF	PlantCLEF2022	94.12 ± 0.02	90.47 ± 0.03	87.43 ± 0.04
	ViT-IN21k	ImageNet21k	97.61 ± 0.01	94.42 ± 0.01	93.06 ± 0.01
	ViT-IN21k-AM (ours)	ImageNet21k	**98.61 ± 0.00**	**94.91 ± 0.01**	**94.17 ± 0.01**

**Table 4 animals-13-03588-t004:** Ablation study with different loss functions on Testing Dataset 1 (TD1). √ indicates the adoption of this loss function.

Method	Cross-Entropy	AM-Softmax	CSA	AUROC	OSCR
RN50-IN1k	√		89.16 ± 0.03	85.11 ± 0.03	79.20 ± 0.03
		√	93.61 ± 0.02	90.84 ± 0.03	87.22 ± 0.02
ViT-MAE	√		77.71 ± 0.03	77.37 ± 0.05	65.92 ± 0.04
		√	82.21 ± 0.04	77.70 ± 0.04	67.74 ± 0.03
ViT-PlantCLEF	√		82.56 ± 0.04	82.10 ± 0.03	72.41 ± 0.02
		√	89.48 ± 0.03	85.46 ± 0.03	80.64 ± 0.03
ViT-IN21k	√		93.57 ± 0.02	91.24 ± 0.02	87.35 ± 0.01
		√	94.49 ± 0.02	91.84 ± 0.01	87.85 ± 0.02

**Table 5 animals-13-03588-t005:** Influence of the margin *m* on the AM-Softmax function. For this ablation study, we conducted experiments using Test Dataset 2 (TD2) with 7 cattle as the known classes and 10 cattle as the unknown classes.

Method	*m*	CSA	AUROC	OSCR
RN50-IN1k-AM	0.4	97.31 ± 0.01	93.14 ± 0.01	92.59 ± 0.02
	0.5	97.55 ± 0.01	93.96 ± 0.01	92.67 ± 0.02
	0.6	97.01 ± 0.01	93.75 ± 0.01	92.75 ± 0.01

**Table 6 animals-13-03588-t006:** Comparison results on closed-set cattle’s face recognition (17 cattle) with our CFOSR method and the generic method. The comparative experiments were conducted based on the more-challenging Testing Dataset 1 (TD1). √ indicates the adoption of this loss function. The boldface represents the best results for the specific evaluation metrics.

Model	Cross-Entropy	ARPL	AM-Softmax	Acc	F1-Score
RN50-IN1k	√			88.80 ± 0.02	88.58 ± 0.02
RN50-IN1k-AM		√	√	93.43 ± 0.01	93.49 ± 0.01
ViT-IN21k	√			92.62 ± 0.01	92.50 ± 0.01
ViT-IN21k-AM		√	√	94.46 ± 0.01	94.53 ± 0.01

**Table 7 animals-13-03588-t007:** Analyzing the metrics including the model Parameters (Params), Floating-Point Operations (FLOPs), training time measured in hours (h), and frames per second (fps) in inference time, we found that the ViT models boasted a larger parameter count and necessitated more time for training. Despite these factors, the ViT models maintained real-time fps values, primarily due to the straightforward nature of the image classification. ↓ indicates that a smaller value is better, while ↑ indicates that a larger value is better.

Model	Params (M) ↓	FLOPs (G) ↓	Training Time (h) ↓	fps ↑
ResNet50	23.533	4.109	2.13	173.51
large ViT	311.296	61.603	17.75	80.36

## Data Availability

The data presented in this study are available upon request from the corresponding author. The data are not publicly available due to project restrictions.

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
