# Peer review of "Improving Known–Unknown Cattle’s Face Recognition for Smart Livestock Farm Management"

_animals, 2023, doi:10.3390/ani13223588_

Round 1

Reviewer 1 Report

Comments and Suggestions for Authors

The work is relevant and can find application in practical use. This work is just the beginning of a large amount of future research in the field of detection and non-contact monitoring of animals. The authors used several methods for detecting animal faces as a way to personally identify the animal. The result of the research was aimed at increasing the accuracy of classification of data volumes. However, it is necessary to finalize the article in accordance with the following recommendations:

Point 1: In paragraph “2.3. Datasets Collection and Preprocessing" technical characteristics of the cameras were not given. It is necessary to describe the technologies on which the cameras are based, the manufacturer of the commercial camera used and its measurement uncertainty. Please indicate if 3D cameras were used.

Point 2: In paragraph “2.3. Datasets Collection and Preprocessing", line 154 there is no explanation for choosing the camera operating mode. Why exactly 15 fps?

Point 3: In paragraph “2.3. Datasets Collection and Preprocessing" it is necessary to provide a schematic representation of how the camera was installed relative to the animals, indicating the characteristics (height, distance, angle of inclination).

Point 4: In paragraph “2.3. Datasets Collection and Preprocessing" it’s not clear how you compared the number of the animal and its number and position in the picture.

Point 5: The work should describe the options for head position relative to the camera and the chance of successful detection. For example, “when recognizing 40% of a face as an effective surface, the chance of successful detection by such and such a method is 60%”.

Point 6: It is worth adding an active link or doi to the reference list.

Reviewer 2 Report

Comments and Suggestions for Authors

I thought this was a well written paper that addresses an important practical issue with considering computer vision (CV) systems for cattle . The paper compared a number of computer architectures and training methods to address the open set classification problem for individual cattle, which would likely be a problem when employing CV systems on farms.

By far the most significant performance improvement was achieved by training CV architectures with general image data-sets and then performing transfer learning with the cattle images. This significant improvement with transfer learning might be indicative that cattle training data-sets were quite small; this could easily be remedied in real CV systems by collecting more images over longer periods.

However, a small performance improvement was also obtained by using open set classification methods to learn more compact class distributions that enable new 'unseen" cattle to be distinguished.  

The following are my recommended changes to improve the paper:

(a) Given there are diffrerent types of farming systems for cattle (feedlot, grazing), it would be useful to specify the type of system this model was tested with (it appears it would be feedlot) and what types of systems this computer vision approach is applicable to in the introduction i.e. I could imagine it would be difficult to use computer vision in a open rangeland, grazing based system.

(b) R is not defined in (1). Furthermore, it is not specified in the experiment section. Is this a fixed margin parameter, please clarify in the paper.

© How are the AMC and margin softmax fns used for training? Figure 1 makes it look like they might be combined in a joint cost function, but I think the results and my own intuition suggests they would be utilised independently as alternate training approaches? Could this be clarified in the paper.

(d) A quick sensitivity analysis of margin m (just a relatively small set of different m value) for margin softmax in this experiment should be provided to understand the impact of margin specification on classification performance? Also if R is a margin for AMC, it should also be performed for this metric.

(e) I think using the openess metric from (3) makes it more difficult to interpret Figure 8. Given the x-axis of Figure 8 is just the number of individual cattle used to train the model (given a fixed number of unknown test cattle), imo, it would make it clearer to use this on the x-axis as opposed to the openess metric to interpet results.

(f)Figure 5 shows distributions between known and unknown class of individuals. Its not clear to me what the negative distance is, more clarity about what metric these distributions are being compared  is required to understand the significance of the result.

(g) Its commenable that the author’s clearly state the limitation of the current approach in the discussion, that is, they can classify unknown individuals as belonging to the unknown class, however, not differentiate between individuals within this class. It would be interesting for the authors to provide some insights into potential ways they may address this individual differentitation issue.

Reviewer 3 Report

Comments and Suggestions for Authors

Dear authors,

After carefully reading your interesting paper on facial recognition in cattle, I have but a few suggestions related mainly to the structure of your manuscript:

- Lines 96-113: No need to present contributions of your study to this section, please move these sentences to the Discussion section and end your Introduction section with the aim(s);

- Lines 115-132: Challenges faced do not need to be presented in the Materials and Method section, they should be moved wither to the introduction section or to the Dissucssions;

- I fail to understand why the authors decided to present two sections of Methods (sub-section 3 and 4 in the manuscript), this makes the manuscript harder to read and it looks like the article was written in a rush. It would be a shame for such important work to leave the impression of sloppy;

- Please emphasize on the accuracy data of your method in both the abstract and conclusions sections.

Round 2

Reviewer 2 Report

Comments and Suggestions for Authors

All revisions have been addressed by the authors

Comments on the Quality of English Language

The English is of a reasonable quality and sufficient for the paper to clearly understood.